# Allergy Test Dosage of Fluorescein Detects Diabetic Retinopathy Changes in Fundus Fluorescein Angiography

**DOI:** 10.3390/diagnostics13233519

**Published:** 2023-11-23

**Authors:** Jian-Feng Yang, Yun Wang, Lingling Zhou, Shaoying Tan, Zhanchi Hu, Tsz Kin Ng, Ling-Ping Cen

**Affiliations:** 1Joint Shantou International Eye Center of Shantou University and The Chinese University of Hong Kong, Shantou 515041, China; yjf@jsiec.org (J.-F.Y.); wy@jsiec.org (Y.W.); zhoull@jsiec.org (L.Z.); shaoying.tan@polyu.edu.hk (S.T.); huzhanchi@aliyun.com (Z.H.); micntk@hotmail.com (T.K.N.); 2School of Optometry, The Hong Kong Polytechnic University, Hong Kong SAR, China; 3Research Centre for SHARP Vision (RCSV), The Hong Kong Polytechnic University, Hong Kong SAR, China; 4Shantou University Medical College, Shantou 515041, China; 5Department of Ophthalmology and Visual Sciences, The Chinese University of Hong Kong, Hong Kong SAR, China

**Keywords:** diabetic retinopathy, fundus fluorescein angiography, fluorescein sodium allergy test

## Abstract

Background: The aims of this study were to evaluate the feasibility of allergy test dosage of fluorescein sodium (1%) for Diabetic Retinopathy (DR) detection in Fundus Fluorescein Angiography (FFA) examination as compared to the regular dosage (20%). Methods: Totally 77 eyes from 42 DR patients were included in this prospective study. Capillary non-perfusion area, neovascularization, diabetic macular edema and microaneurysms were measured by FFA and compared at 1, 5 and 15 min after intravenous injection of 1% or 20% fluorescein sodium. Results: There was no statistically significant difference in the proportions of capillary non-perfusion area and diabetic macular edema as well as the amount of neovascularization between the 1% and 20% fluorescein sodium groups. Yet, the 1% group had a significantly a smaller number of microaneurysms than the 20% group at 1 min (*p* < 0.001) and a smaller number of eyes with diabetic macular edema than the 20% group at 5 (*p* = 0.032) and 15 min (*p* = 0.015). The images from patients with clear vitreous had better quality than the images from patients with vitreous opacity (all *p* < 0.05, except comparison on neovascularization at 5 min: *p* > 0.999). All examined indexes showed high correlations between the 1% and 20% groups (r > 0.8, *p* < 0.001). Conclusions: This study demonstrated that 1% fluorescein sodium could detect the changes of DR comparably to the regular dosage.

## 1. Introduction

Diabetic Retinopathy (DR), the most common vascular disease affecting the retina, is a leading cause of irreversible blindness and visual impairment in the working-age population [1]. The incidence of DR is not related to gender or age, but positively correlated with the course of diabetes [2,3]. The incidence of DR is even higher when combined with hypertension and hyperlipidemia, whereas blood pressure control can alleviate the progression of DR [4,5].

Compared to fundus photography, Fundus Fluorescein Angiography (FFA) has greatly improved the screening and diagnosis of DR. FFA is the gold standard for the clinical evaluation of morphology and pathophysiology of retinal vasculature [6,7], including microaneurysms, microvascular abnormalities and neovascularization as well as for the detection of non-perfused areas in the retina [8,9]. An Early Treatment Diabetic Retinopathy Study (ETDRS) reported that fluorescein leakage, capillary loss and dilatation, and that arteriolar abnormalities are associated with DR severity and the likelihood of progression to proliferative retinopathy, whereas the severity of fluorescein leakage is also associated with macular edema [10]. However, adverse events due to the allergic reactions to sodium fluorescein have been reported during the FFA examination, including nausea and vomiting [11,12], although the majority of the adverse reactions are mild [13].

To avoid adverse allergic reactions, the patients are required to perform a fluorescein sodium allergy test with the injection of 1% fluorescein sodium before the actual FFA examination. If the fluorescein sodium allergy test is positive, the patient will not receive the FFA examination, and the DR-related fundus lesions cannot be accurately evaluated. Individuals especially with previous history of allergies and hypertension are at high risk [14,15,16]. Low concentration of fluorescein sodium has been used for endoscopic confocal laser microscopy on gastric mucosa in patients with intestinal metaplasia [17]. Clear confocal microendoscopic images could be obtained by injecting 0.02 mL/kg fluorescein sodium and effectively diagnosing intestinal metaplasia in gastric mucosa. In this study, we aimed to evaluate the feasibility of allergy test dosage of fluorescein sodium (1%) for detecting the DR-related changes in FFA examination compared to the regular dosage (20%).

## 2. Materials and Methods

### 2.1. Study Subjects

Total 42 study subjects diagnosed with DR were recruited at the Joint Shantou International Eye Center of Shantou University and the Chinese University of Hong Kong from March to April 2018. For the DR diagnosis, all patients received complete ophthalmic examinations, including visual acuity test, intraocular pressure measurement, slit-lamp, fundus and FFA examination, and fundus photography. The history of diabetes was also recorded.

The diagnosis of DR is based on the diabetic retinopathy preferred practice pattern^®^ guidelines by the American Academy of Ophthalmology (2017) [18]. The interval of FFA examination and fundus photography of all study subjects was less than two weeks. Fundus photography was taken by the Topcon Fundus Camera and Image Processing System TRC-50DX (TOPCON, Tokyo, Japan). For the FFA examination, one or a plurality of conditions of the capillary non-perfusion area, neovascularization, microaneurysms and diabetic macular edema in the posterior retina were detected. The patients suffering from other fundus diseases, such as retinal venous occlusion, age-related macular degeneration, retinal arterial occlusion and high myopia chorioretinopathy, were not included in this study. The eyes were excluded if the fundus could not be detected. The study protocol was approved by the Ethics Committee for Human Research at the Joint Shantou International Eye Center of Shantou University and the Chinese University of Hong Kong, which is in accordance with the tenets of the Declaration of Helsinki. Written informed consent was obtained from all study subjects after explanation of the nature and possible consequences of the study.

In addition, electronic medical records of the patients received fluorescein sodium allergy tests and FFA examinations in our hospital from January 2016 to December 2017 were retrieved to estimate the incidence rate of fluorescein sodium allergy tests and the adverse reactions during FFA examinations.

### 2.2. Fundus Fluorescein Angiography

All patients first received the injection of 5 mL of 1% fluorescein sodium followed by an immediate FFA examination (1% fluorescein sodium group), and then received the injection of 2.5 mL of 20% fluorescein sodium followed by another immediate FFA examination (20% fluorescein sodium group). All injections used the same batch (160901) and the same specifications (3 mL: 0.6 g) of fluorescein sodium by the same manufacturer (Guangzhou Baiyunshan Mingxing Pharmaceutical Co. Ltd., Guangzhou, China). The FFA images were taken by the Spectralis Heidelberg Retina Angiography (HRA) imaging platform (Heidelberg Engineering, Heidelberg, Germany) or Spectralis HRA + OCT (Heidelberg Engineering). The FFA images were taken at 1, 5 and 15 min after fluorescein sodium injection using the Spectralis HRA mode. According to FFA and fundus photography, the images were divided into vitreous opacified and clear groups, and the eyes with obvious vitreous opacity would be assigned to the vitreous opacified group. Eyes with severe vitreous opacity, making it impossible to view the fundus, were excluded. In contrast, eyes without obvious vitreous opacity would be assigned to the vitreous clear group.

### 2.3. Fundus Fluorescein Angiography Image Analysis

Standard fluorescein angiograms were analyzed quantitatively and qualitatively by a single examiner (J.F.Y.). The examiner was masked that each image could not be paired up with the corresponding patient. The features obtained from the FFA images included the capillary non-perfusion area, neovascularization, diabetic macular edema and microaneurysms. The numbers of neovascularization and microaneurysms were counted manually (Figure 1K–N). The capillary non-perfusion area and diabetic macular edema were measured by the Image J software (version 1.8.0; National Institute of Health, Bethesda, MD, USA; Figure 1A,B,I,J). The effects of hemorrhage and weak fluorescence caused by cotton wool spots were excluded. The area ratios of the capillary non-perfusion area or the diabetic macular edema area were calculated as the ratio of the measured capillary non-perfusion area or diabetic macular edema area to the total area of an image.

### 2.4. Statistical Analysis

The results were presented as mean ± standard deviation (SD) or proportions. Paired *t*-test or Wilcoxon test was used to compare the differences between the two tested dosages. Pearson or Spearman analysis was chosen to evaluate the correlation between the two tested dosages. The χ^2^ test was adopted to analyze the differences in the number of eyes with the examined indexes between the two dosage groups, and the differences in the image quality between the vitreous opacified and clear groups. *p* < 0.05 was considered as statistically significant. All statistical analyses were performed using commercially available software (SPSS, version 17.0; SPSS Inc., Chicago, IL, USA).

## 3. Results

### 3.1. Patient Demographics

A total of 77 eyes (38 left and 39 right eyes) from 42 patients (22 male and 20 female subjects) diagnosed with DR were included in this study, with an overall mean age of 55.2 ± 9.7 (range: 30–75) years old (Table 1). Forty-six eyes showed no obvious vitreous opacity, and 31 eyes showed obvious vitreous opacity. Non-proliferative DR was found in 24 eyes, and proliferative DR in 53 eyes. There was no significant difference in age (*p* = 0.492), gender (*p* = 0.611) and laterality (*p* = 0.429) between the vitreous opacified and clear groups. Yet, the vitreous opacified group had a higher proportion of eyes with proliferative DR (80.6%) than the vitreous clear group (60.9%, *p* = 0.066).

### 3.2. Adverse Reactions to Fluorescein Sodium

From January 2016 to December 2017, 2864 patients received FFA examination in our hospital, and 42 patients showed positive in the fluorescein sodium allergy test with an incidence rate of 1.47%, including 8 DR patients (19.05%). 171 patients were negative in the fluorescein sodium allergy test but showed adverse reactions during FFA examination with an incidence rate of 5.98%. Most of the adverse reactions were mild.

In this study, none of the 42 study subjects showed positive in the fluorescein sodium allergy test, yet 3 (7.1%) of the patients showed adverse reactions to the injection of 20% fluorescein sodium. (1) The first patient showed a rash on the right side of the face at 10 min after 20% fluorescein sodium injection, and the rash subsided after drinking plenty of water; (2) The second patient felt flustered and nausea at 1 min after 20% fluorescein sodium injection, and the symptoms disappeared after deep breathing; 3) The third patient also felt nausea at 1 min after 20% fluorescein sodium injection, and the symptoms relieved after deep breathing.

### 3.3. Outcomes of 1% Fluorescein Sodium Injection in Fundus Fluorescein Angiography

FFA imaging showed that the fluorescence perfusion with 1% fluorescein sodium injection was lower than that of the 20% dosage; however, microaneurysms, capillary non-perfusion area, neovascularization, and diabetic macular edema could still be clearly observed (Figure 1A–F). The fluorescence subsided in the 1% dosage group was more pronounced than that in the 20% dosage group, especially at 15 min after injection.

### 3.4. Capillary Non-Perfusion Area

Among the 77 eyes diagnosed with DR, posterior pole capillary non-perfusion area was detected in 49 (63.6%) of the eyes in the 20% fluorescein sodium group and just 40 (51.9%) of the eyes in the 1% dosage group at 1 min after injection (Figure 1A,B), but there was no significant difference between the 2 dosage groups (*p* = 0.236; Table 2). At 5 min after injection, posterior pole capillary non-perfusion area could be observed in 30 (39.0%) of the eyes in the 1% dosage group (Figure 1C,D), which showed a significant difference compared to the 49 (63.6%) eyes detected in the 20% dosage group (*p* = 0.038). Nevertheless, there was no statistically significant difference in the capillary non-perfusion area ratio between the 1% and 20% dosage groups at 1 (1%: 5.91 ± 7.98% and 20%: 6.07 ± 8.11%, *p* = 0.179) and 5 min (1%: 5.64 ± 7.81% and 20%: 5.65 ± 7.80%, *p* = 0.903). The capillary non-perfusion area ratios of the 1% dosage group were positively correlated with that of 20% dosage at 1 (r = 0.993, *p* < 0.001; Figure 2A) and 5 min (r = 0.998, *p* < 0.001; Figure 2B).

### 3.5. Neovascularization

Among the 77 eyes diagnosed with DR, posterior pole neovascularization was detected in 38 (49.4%) of the eyes in the 20% dosage group and 38 eyes (49.4%) in the 1% dosage group at 5 min after injection (*p* > 0.999; Table 2 and Figure 1C,D). At 15 min after injection, neovascularization could only be detected in 33 (42.9%) of the eyes in the 1% dosage group (Figure 1E,F). However, there was no significant difference compared to the 38 (49.4%) of the eyes in the 20% dosage group (*p* = 0.556). Moreover, there was also no statistically significant difference in the number of posterior pole neovascularization between the 1% and 20% dosage groups at 5 (1%: 2.95 ± 1.87 and 20%: 3.05 ± 1.90, *p* = 0.102) and 15 min after injection (1%: 2.55 ± 1.48 and 20%: 2.67 ± 1.56, *p* = 0.102; Table 2). There was a positive correlation between the 1% and 20% dosage groups at 5 (r = 0.984, *p* < 0.001; Figure 2E) and 15 min (r = 0.977; *p* < 0.001; Figure 2F).

### 3.6. Diabetic Macular Edema

Among the 77 eyes diagnosed with DR, diabetic macular edema was detected in 33 (42.9%) of the eyes in the 20% dosage group, but in just 23 (29.9%) of the eyes in the 1% dosage group at 5 min after injection (*p* = 0.032; Table 2 and Figure 1C,D). At 15 min after injection, diabetic macular edema could only be detected in 20 (26.0%) of the eyes in the 1% dosage group (Figure 1E,F), as compared to the 33 (42.9%) of the eyes in the 20% dosage group (*p* = 0.015). Yet, there was no statistically significant difference in the diabetic macular edema area ratio between the 1% and 20% dosage groups at 5 (1%: 1.79 ± 1.01% and 20%: 1.79 ± 0.99%, *p* = 0.932) and 15 min after injection (1%: 2.57 ± 1.71% and 20%: 2.66 ± 1.75%, *p* = 0.062; Table 2). The diabetic macular edema area ratios of the 1% dosage group were positively correlated with that of the 20% dosage group at 5 (r = 0.992, *p* < 0.001; Figure 2C) and 15 min (r = 0.994, *p* < 0.001; Figure 2D).

### 3.7. Microaneurysms

At 1 min after fluorescein sodium injection, microaneurysms could be detected in 77 (100.0%) of the eyes in the 20% dosage group but just 55 eyes (71.4%) in the 1% dosage group at 1 min after injection (*p* < 0.001; Table 2 and Figure 1K,L). Moreover, the number of microaneurysms was significantly lesser in the 1% dosage group (182.5 ± 139.1) than the 20% dosage group (240.5 ± 138.4, *p* < 0.001; Table 2). Yet, there was a positive correlation between the number of microaneurysms in the 1% and 20% dosage group groups (r = 0.823, *p* < 0.001; Figure 2G).

### 3.8. Image Quality in Patients with Vitreous Opacity

For the capillary non-perfusion area, the number of clear images in the vitreous opacified group was significantly lower than that in the vitreous clear group at 1 (58.82% and 93.75%, respectively, *p* = 0.009) and 5 min after injection (35.29% and 75.00%, respectively, *p* = 0.007, Table 3 and Figure 1O,P). Similarly, for the microaneurysms, the number of clear images in the vitreous opacified group was significantly lower than that in the vitreous clear group at 1 min after injection (48.39% and 86.96%, respectively, *p* < 0.001). For diabetic macular edema, the number of clear images in the vitreous opacified group was significantly lower than that in the vitreous clear group at 5 (36.36% and 90.48%, respectively, *p* = 0.001) and 15 min after injection (33.33% and 76.19%, respectively, *p* = 0.015). For neovascularization, the number of clear images in the vitreous opacified group was significantly lower than that in the vitreous clear group at 15 min (69.23% and 96.00%, respectively, *p* = 0.021). However, for neovascularization at 5 min after injection, there were no significant differences in the number of clear images between the vitreous opacified and the vitreous clear groups.

## 4. Discussion

Results from this study in DR patients who completed the FFA examinations in both 1% and 20% fluorescein sodium dosages showed that (1) there was no statistically significant difference in capillary non-perfusion area between the 1% and 20% fluorescein sodium groups despite less number of eyes detected with the posterior pole capillary non-perfusion area in the 1% dosage group; (2) there was no statistically significant difference in the number of posterior pole neovascularization and the number of eyes with neovascularization between the 1% and 20% dosage groups; (3) there was no statistically significant difference in the diabetic macular edema area ratio between the 1% and 20% dosage groups although less eyes could be detected with diabetic macular edema in the 1% dosage group; (4) the number of microaneurysms detected was lesser in the 1% dosage group as compared to the 20% dosage group. Collectively, our results suggested the capability of 1% fluorescein sodium detecting capillary non-perfusion area, neovascularization, and some extent of diabetic macular edema and microaneurysms in FFA examination for DR evaluations.

Adverse allergic reactions to 20% sodium fluorescein have been frequently reported during the FFA examination. The rate of adverse reactions varies in different countries: 9.72% in Recife (1039 patients) [14], 7.5% in Florence, Italy (6524 patients) [15], and 11.2% in Liverpool, the United Kingdom (358 patients) [16]. In this study, the rate of adverse reactions was 7.1%. For the individuals with adverse reactions, the FFA images usually could not be obtained for DR evaluations and treatment recommendations. However, if the FFA imaging is crucial for further management decisions, the application of 1% fluorescein sodium could be an option to detect the DR-related changes, especially for the patients with an allergy history or at high risk of adverse reactions. Moreover, since fluorescein is excreted in the urine relatively unchanged within 24–36 h after administration [19,20], 1% fluorescein (easier to be excreted) would be safer than 20% fluorescein for DR patients with concurrent nephropathy.

The application of a lower concentration of fluorescein sodium has been previously evaluated. A previous study compared the injection of 3 mL of 25% with 5 mL of 10% fluorescein sodium solution in normal volunteers and patients with diverse ophthalmic disorders. The study demonstrated that the injection of 25% fluorescein sodium has better visualization, angiogram quality and 5 min phase angiogram than the 10% injection [21]. However, the most common untoward reactions of mild, transient nausea could be observed in both the 10% and 25% groups, and there was no significant difference in the incidence and severity of adverse reactions between the 2 tested dosages. In this study, instead of 10% fluorescein sodium, we compared the injection of 1% fluorescein sodium with the regularly applied concentration of 20%. Among the study subjects, no adverse reaction was found under the application of 1% fluorescein sodium, yet 3 events of adverse reaction were reported with the injection of 20% fluorescein sodium, suggesting that injection of 1% fluorescein sodium has lower chance of inducing adverse reactions than the 20% fluorescein sodium.

In the current study, we compared the feasibility of both dosages of fluorescein sodium in detecting the retinal features in DR patients with or without vitreous opacity. In the vitreous clear group, our results indicated that the injection of 1% fluorescein sodium was able to detect the DR-related features effectively in the FFA examination. High-resolution images could be obtained from the patients injected with 1% fluorescein sodium, which is similar to those injected with 20% fluorescein sodium. No significant difference was observed in detecting capillary non-perfusion area, diabetic macular edema and neovascularization. However, in the patients with vitreous opacity, the images of the 1% fluorescein sodium group were greatly affected (Figure 1). Our results show that the eyes with vitreous opacity had significantly a smaller number of clear FFA images than the eyes with clear vitreous (Table 3 and Figure 1O,P), suggesting the vitreous opacity could affect the feasibility of 1% fluorescein sodium in FFA examination. For the patients with vitreous opacity, it is difficult to focus accurately to obtain clear FFA images with 1% fluorescein sodium injection. On the contrary, stronger black-and-white contrast in the images with the 20% fluorescein sodium injection could show a better focus.

With the advancement in technology, Optical Coherence Tomography Angiography (OCTA), as a new technology, could also visualize most of the DR-related vascular changes, including microaneurysms, capillary non-perfusion area, retinal microvascular abnormalities and neovascularization [22,23]. However, the area of OCTA for a scan was limited. It could not allow a large-area scan of retina. The larger the scanning range, the worse the imaging effect was, and the better the patient cooperation needed is. In addition, OCTA also showed sensitivity to motion artifact and inability to see leakage [24,25]. And FFA was more sensitive in identifying microaneurysms than OCTA [26]. Therefore, FFA was still an important strategy for comprehensive DR evaluations, and low dose of fluorescein sodium could be an option for specific situations mentioned above.

## 5. Limitations

Though two FFA examinations with 1% and 20% fluorescein sodium were conducted by the same experienced examiner, we could not ensure the images of the 2 dosage groups were taken in the exact same positions. Therefore, only the post-pole retinal images with optic disc and macula were used for the comparisons in this study. To obtain reliable results of comparisons, overlapping the images from the two dosages of the same patient was implemented according to the position of the optic disc and retinal blood vessels, and only the overlapping part of the two images was selected for comparison (Figure 1G,H).

## 6. Conclusions

In summary, this study revealed that 1% fluorescein sodium showed adequate feasibility in detecting capillary non-perfusion area, neovascularization, and some extent of diabetic macular edema and microaneurysms in FFA examination for DR patients. FFA examination with 1% fluorescein sodium could be a considerable option in the following situations: (1) Fundus could be observed clearly without obvious vitreous opacity. (2) For specific patients, acquiring comprehensive FFA images is crucial for DR evaluations and treatment recommendations. (3) Patients with an allergy history or at high risk of adverse reactions to regular dose of fluorescein sodium (20%) in FFA examination.

## Figures and Tables

**Figure 1 diagnostics-13-03519-f001:**
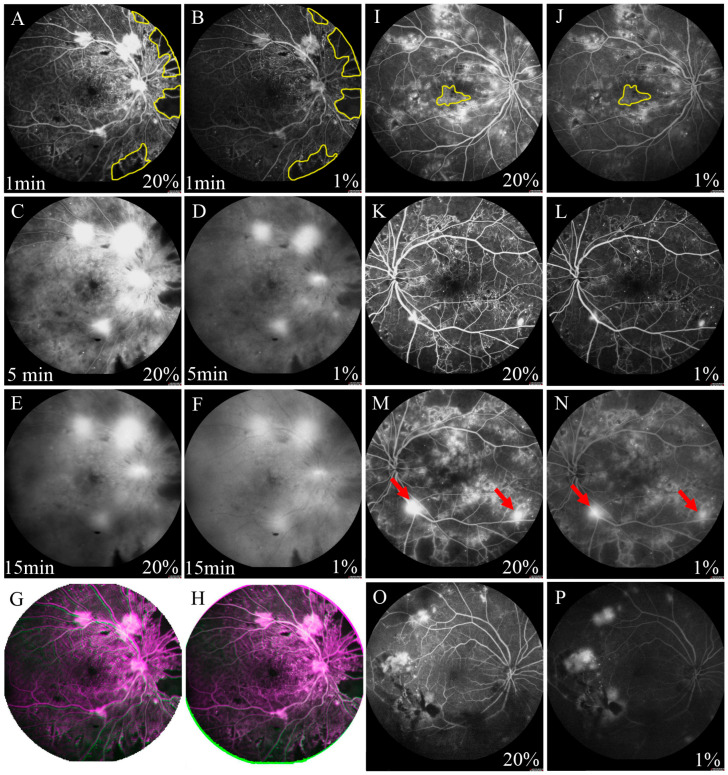
Comparison of FFA images with 2 dosages of fluorescein sodium. (**A**–**F**) showed the FFA images of the same patient. A: The 20% dosage group at 1 min; (**B**): The 1% dosage group at 1 min; (**C**): The 20% dosage group at 5 min; (**D**): The 1% dosage group at 5 min; (**E**): The 20% dosage group at 15 min; (**F**): The 1% dosage group at 15 min. (**A**,**B**): Capillary non-perfusion area was measured by the Image J software (the yellow area). (**G**,**H**): Overlapped the images from 2 dosages according to the position of the optic disc and retinal blood vessels, and selected the overlap part. The green and pink part in the surrounding area were removed. (**I**,**J**): Diabetic macular edema was measured by the Image J software (the yellow area). (**K**,**L**): The numbers of microaneurysms were counted manually. (**M**,**N**): The numbers of neovascularization (arrow) were counted manually. (**O**,**P**): Vitreous opacified group with 20% and 1% fluorescein sodium.

**Figure 2 diagnostics-13-03519-f002:**
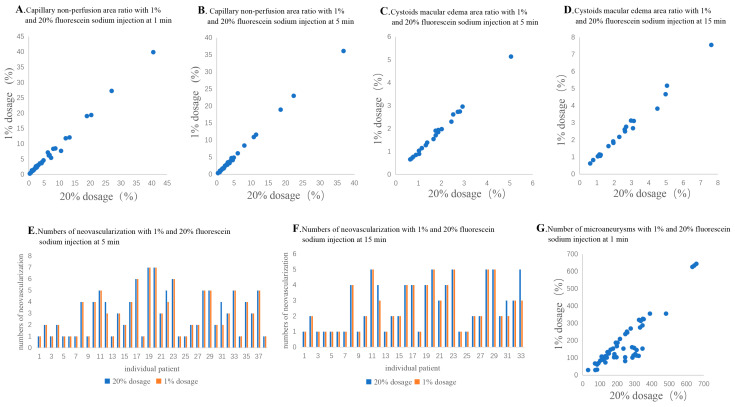
The correlation of examined indexes. (**A**,**B**): The correlation of capillary non-perfusion area ratio between the 1% and 20% dosage groups at 1 min (**A**) and 5 min (**B**); (**C**,**D**): The correlation of diabetic macular edema area ratio between the 1% and 20% dosage groups at 5 min (**C**) and 15 min (**D**); (**E**,**F**): The correlation of neovascularization numbers between the 1% and 20% dosage groups at 5 min (**E**) and 15 min (**F**); (**G**): The correlation of microaneurysms numbers between the 1% and 20% dosage groups at 1 min.

**Table 1 diagnostics-13-03519-t001:** Demographics of diabetic retinopathy study subjects with or without vitreous opacity.

	Total	Vitreous Opacified	Vitreous Clear	*p*
*n* (eyes)	77	31	46	/
Age (mean ± SD; years)	55.23 ± 9.70	56.42 ± 9.96	54.43 ± 9.55	0.492
Gender (male/female)	42/35	18/13	24/22	0.611
Laterality (OD/OS)	39/38	14/17	25/21	0.429
NPDR/PDR	24/53	6/25	18/28	0.066

*n*: number; NPDR: non-proliferative diabetic retinopathy; OD: right eye; OS: left eye; PDR: proliferative diabetic retinopathy; SD: standard deviation.

**Table 2 diagnostics-13-03519-t002:** Fundus fluorescein angiography examinations with 1% and 20% fluorescein sodium injection.

		1% Fluorescein Sodium	20% Fluorescein Sodium	Paired Differences	*p*
**Capillary non-perfusion area**					
1 min	Eyes (*n*; Y/N)	40/37	49/28		0.236
	Ratio	5.91 ± 7.98%	6.07 ± 8.11%	0.0016 ± 0.0061	0.179
5 min	Eyes (*n*; Y/N)	30/47	49/28		0.038
	Ratio	5.64 ± 7.81%	5.65 ± 7.80%	0.0001 ± 0.0028	0.903
**Neovascularization**					
5 min	Eyes (*n*; Y/N)	38/39	38/39		>0.999
	Number	2.95 ± 1.87	3.05 ± 1.90	0.1053 ± 0.3883	0.102
15 min	Eyes (*n*; Y/N)	33/44	38/39		0.556
	Number	2.55 ±1.48	2.67 ± 1.56	0.1212 ± 0.4152	0.102
**Cystoid macular edema**					
5 min	Eyes (*n*; Y/N)	23/54	33/44		0.032
	Ratio	1.79 ± 1.01%	1.79 ± 0.99%	0.0000 ± 0.0007	0.932
15 min	Eyes (*n*; Y/N)	20/57	33/44		0.015
	Ratio	2.57 ± 1.71%	2.66 ± 1.75%	0.0008 ± 0.0019	0.062
**Microaneurysms**					
1 min	Eyes (*n*; Y/N)	55/22	77/0		<0.001
	Number	182.45 ± 139.11	240.51 ± 138.42	58.05 ± 63.55	<0.001

*n*: number of eyes; Y: yes; N: no.

**Table 3 diagnostics-13-03519-t003:** The ratio of clear 1% fundus fluorescein angiography imaging in eyes with or without vitreous opacity.

		Vitreous Opacified *n* = 31	Vitreous Clear *n* = 46	*p*
Capillary non-perfusion area (*n*, %)	1 min	10/17 (58.82%)	30/32 (93.75%)	0.009
	5 min	6/17(35.29%)	24/32 (75.00%)	0.007
Neovascularization (*n*, %)	5 min	13/13 (100.00%)	25/25 (100.00%)	>0.999
	15 min	9/13 (69.23%)	24/25 (96.00%)	0.021
Cystoid macular edema (*n*, %)	5 min	4/11(36.36%)	19/21 (90.48%)	0.001
	15 min	4/12 (33.33%)	16/21 (76.19%)	0.015
Microaneurysms (*n*, %)	1 min	15/31 (48.39%)	40/46 (86.96%)	<0.001

*n*: number of eyes.

## Data Availability

The data presented in this study are available in articles.

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
