# Peer review of "Allergy Test Dosage of Fluorescein Detects Diabetic Retinopathy Changes in Fundus Fluorescein Angiography"

_diagnostics, 2023, doi:10.3390/diagnostics13233519_

Round 1
Reviewer 1 Report
Comments and Suggestions for Authors
I evaluate the paper diagnostics-2575590-peer-review
It is an interesting study, good ideas to compare the concentration of Fluorescein
The introduction. Good. Comprehensive. It should be completed with studies that used 1% fluorescein and if they had results.
Material and Methods. Good described procedures. I don't understand the comparison between those with vitreous opacities and those without. A condition of this procedure so that it can be interpreted correctly is the clarity of the vitreous.
Results. It is interesting results. Aspected results about comparative groups. It is interesting how the percentage of neovessels identified was higher in the group with vitreous opacities vs. without.
Figure 2. The graphs are not very understandable. They should be redone, to be clearer.
Diabetic macular edema. 1% vs 20% better results. Also better on microaneurysms 20% vs 1%
Discussions. In truth, there are no statistically significant differences, but as a number of patients identified with diabetic macular edema, which has an important role in the decrease of visual acuity, the scale tilts towards 20%
Could you explain more clearly what the major advantages of the 1% concentration would be and to which group of patients it is addressed
Why would we do an angiofluorography in those with vitreous opacities. Compared to OCT, I don't think it is an advantage.
Conclusions. Can be improved. The only one that seems important to me: Patients with allergy history or at high risk of adverse reactions to regular dose of fluorescein sodium (20%) in FFA examination.
Good luck
Kind regards
Author Response
Reviewer 1:
1. The introduction. Good. Comprehensive. It should be completed with studies that used 1% fluorescein and if they had results.
Response: Thank you for the suggestion. 1% fluorescein is used for allergy test,but no study with comparison of 1% and 20%.We mentioned a study about low concentration of fluorescein sodium has used for the endoscopic confocal laser microscopy on gastric mucosa in patients with intestinal metaplasia(page 2).Also a two-years review of our hospital is mentioned,which is about 1% fluorescein.( page 4)
- Material and Methods. Good described procedures. I don't understand the comparison between those with vitreous opacities and those without. A condition of this procedure so that it can be interpreted correctly is the clarity of the vitreous.
Response: Thank you for the suggestion. We mentioned exclusion criteria that the eyes were excluded if the fundus could not be detected(page 2).So,if vitreous opacities caused the fundus could not be detected, the eyes were excluded. The eyes with obvious vitreous opacity would be assigned into the vitreous opacified group. In contrast, the eyes without obvious vitreous opacity would be assigned into the vitreous clear group.(page 3)
3.Results. It is interesting results. Aspected results about comparative groups. It is interesting how the percentage of neovessels identified was higher in the group with vitreous opacities vs. without.
Response: Thank you for the comment. However,the percentage of neovessels identified was higher in the group without vitreous opacities vs. with.For neovascularization, the number of clear images in the vitreous group was significantly lower than that in the vitreous clear group at 15 minutes (69.23% and 96.00% respectively, p = 0.021).( page 7)
Figure 2. The graphs are not very understandable. They should be redone, to be clearer.
Response: Thank you for the suggestion. Figure 2 expounded the linear correlation of examined indexes.
Could you explain more clearly what the major advantages of the 1% concentration would be and to which group of patients it is addressed.
Response: Thank you for the question. 1% concentration could accurately evaluate the DR-related fundus lesions if the fluorescein sodium allergy test is positive.FFA examination with 1% fluorescein sodium could be a considerable option in the following situations: (1) Fundus could be observed clearly without obvious vitreous opacity. (2) For specific patients, acquiring comprehensive FFA images are crucial for DR evaluations and treatment recommendations. (3) Patients with allergy history or at high risk of adverse reactions to regular dose of fluorescein sodium (20%) in FFA examination.(page 9)And Low concentration of fluorescein sodium was better for patients with poor renal function.
- Why would we do an angiofluorography in those with vitreous opacities. Compared to OCT, I don't think it is an advantage.
Response: Thank you for the question.Most patients have a little of vitreous opacities,we can do an angiofluorography for them.Moreover, if OCT can not dectect the fundus of patients with vitreous opacities,FFA can dectect DR changes,especially peripheral fundus.
- Conclusions. Can be improved. The only one that seems important to me: Patients with allergy history or at high risk of adverse reactions to regular dose of fluorescein sodium (20%) in FFA examination.
Response: Thank you for the suggestion.But low concentration of fluorescein sodium was better for patients with poor renal function.It may be important for these patients.
Reviewer 2 Report
Comments and Suggestions for Authors
General comments
Fundus fluorescein angiography (FFA) is an established technique for detection of diabetic retinopathy (DR). The dose of fluorescein has a small risk of allergic reactions including anaphylaxis which is regarded as fluorescein dose dependent. Accordingly a pre-test allergy test is required. This paper asks whether the pre-test dose itself might be sufficient to provide diagnostic evidence of DR. The authors find that the small dose of fluorescein is sufficient to detect most features of DR except microaneurysms which are an early clinical sign of DR..
Two parameters are under evaluation in FFA studies (a) the quality of the images; and (b) the diagnostic detection rate for each clinical feature. In this paper, only the second parameter is evaluated. For practical use of the low dose, information on image quality would be helpful.
Feature detection rate appears for be good for both doses of fluorescein apart from microaneurysm detection which was lower with low dose fluorescein. Accordingly the low does FFA may not be useful in screening studies for DR.
The design of the study suffers from the image analysis methodology since only one observer was used. Despite masking, there is no indication of within observer error rate or of interobserver error rate.
A similar paper comparing two doses of fluorescein recruited multiple observers to generate data and then assessed the data using fuzzy logic thus reducing potential observer error (see ref below). For example, evaluation of areas of retinal ischemia is highly subjective and involves considerable variation between observers.
There are a number of systems available for automated detection of the features of DR which would reduce this potential variability.
Specific comments
please check details of reference 10 for correct citation.
What was the time interval between low does and high does FFA?
Fluorescein leakage from vessels persists in the tissues for several hours. This may complicate interpretations with a second FFA.
The degree of vitreous opacity is not described.
Reference
Patel, V. et al. Randomized, Comparative Study of Full- and Half-Dose Fluorescein Angiography. J Vitreoretin Dis 5, 337-344, doi:10.1177/2474126420975310 (2021).
Author Response
- The design of the study suffers from the image analysis methodology since only one observer was used. Despite masking, there is no indication of within observer error rate or of interobserver error rate.
A similar paper comparing two doses of fluorescein recruited multiple observers to generate data and then assessed the data using fuzzy logic thus reducing potential observer error (see ref below). For example, evaluation of areas of retinal ischemia is highly subjective and involves considerable variation between observers.
There are a number of systems available for automated detection of the features of DR which would reduce this potential variability.
Response: Thank you for the suggestion. A similar paper mentioned one of the limitations of their study lies in their interrater reliability. They attributed the lack of consistency between raters to differences in their training in assessing FA studies and the inherent subjectivity of grading the quality of photographs.We had the same problem when we try to analyse data with other observer.So only one observer was used finally.
please check details of reference 10 for correct citation.
Response: Thank you for the suggestion. The reference has been rephrased in the revised manuscript. (page 10).
What was the time interval between low does and high does FFA?
Fluorescein leakage from vessels persists in the tissues for several hours. This may complicate interpretations with a second FFA.
Response: Thank you for the comment. The time interval between low does and high does FFA was over 15min. This may complicate interpretations with a second FFA(20% dose).But over 15min,1% dose has only a little of impact to 20% dose.
The degree of vitreous opacity is not described.
Response: Thank you for the suggestion. The eyes with obvious vitreous opacity would be assigned into the vitreous opacified group. The eyes were excluded if obvious vitreous opacity caused fundus could not be detected. The sentence has been added in the revised manuscript. (page 3)
Round 2
Reviewer 1 Report
Comments and Suggestions for Authors
The authors made the correction!
Methods: Could you write exactly criteria of inclusion in the study, age, dg......?
The results is very nice!
I don t understabdwhy dose with vitreos opacities were taken into acount? it makes viwing dificcult anyway!!!!!
Author Response
Thank you very much for taking the time to review this manuscript. Please find the detailed responses below and the corresponding revisions highlighted in the re-submitted files. We have revised our manuscript accordingly with red highlights.
- Methods: Could you write exactly criteria of inclusion in the study, age, dg......?
Response: Thank you for the suggestion. Study subjects diagnosed as DR were recruited at our hospital from March to April 2018.Patients of any age may be recruited. So our criteria of inclusion didn’t include age. For the DR diagnosis, all patients received complete ophthalmic examinations, including visual acuity test, intraocular pressure measurement, slit-lamp, fundus and FFA examination, and fundus photography. The history of diabetes was also recorded. The diagnosis of DR is based on the diabetic retinopathy preferred practice pattern® guidelines by the American Academy of Ophthalmology (2017). The interval of FFA examination and fundus photography of all study subjects was less than two weeks. For the FFA examination, one or a plurality of conditions of the capillary non-perfusion area, neovascularization, microaneurysms and diabetic macular edema in the posterior retina were detected. The patients suffering from other fundus diseases, such as retinal venous occlusion, age-related macular degeneration, retinal arterial occlusion and high myopia chorioretinopathy, were not included in this study. The eyes were excluded if the fundus could not be detected(page 2). Finally, total 77 eyes (38 left and 39 right eyes) from 42 patients (22 male and 20 female subjects) diagnosed as DR were included in this study, with an overall mean age of 55.2 ± 9.7 (range: 30 – 75) years old (page 4).
- I don’t understand why dose with vitreous opacities were taken into acount? it makes viewing dificcult anyway!!!!!
Response: Thank you for the question. Our results showed that the eyes with vitreous opacity had significantly a smaller number of clear FFA images than the eyes with clear vitreous (Table 3 and Figure 1O and P), suggesting the vitreous opacity could affect the feasibility of 1% fluorescein sodium in FFA examination. FFA examination with 1% fluorescein sodium could be a considerable option if fundus could be observed clearly without obvious vitreous opacity (page 9).
Reviewer 2 Report
Comments and Suggestions for Authors
The authors have resolved a potential inter-observer variation problem by using only only observer which may introduce some bias. But given the circumstances this is not fatal.
Comments on the Quality of English Languagesome of the revisions require editing for Englsh usage.
Author Response
Thank you very much for taking the time to review this manuscript. Please find the detailed responses below and the corresponding revisions highlighted in the re-submitted files. We have revised our manuscript accordingly with red highlights.
Comments on the Quality of English Language:Some of the revisions require editing for English usage.
Response: Thank you for the suggestion. Some sentences and words have been rephrased in the revised manuscript with red highlights.